# Preparing High-Purity Anhydrous ScCl₃ Molten Salt Using One-Step Rapid Heating Process

**Junhui Xiao** [1,2,3,4,*]**, Chao Chen** [1,2]**, Wei Ding** [1]**, Yang Peng** [1]**, Kai Zou** [1]**, Tao Chen** [1] **and Zhiwei Zou** [1]

[1] Sichuan Provincial Engineering Lab of Non-Metallic Mineral Powder Modification and High-Value Utilization, Southwest University of Science and Technology, Mianyang 621010, China; chenchaoswust@hotmail.com (C.C.); dingwei@mails.swust.edu.cn (W.D.); pengyang@mails.swust.edu.cn (Y.P.); zoukaiswust@hotmail.com (K.Z.); chentao@mails.swust.edu.cn (T.C.); zouzhiweiswust@hotmail.com (Z.Z.)

[2] Institute of Multipurpose Utilization of Mineral Resources, Chinese Academy of Geological Sciences, Chengdu 610041, China

[3] Key Laboratory of Sichuan Province for Comprehensive Utilization of Vanadium and Titanium Resources, Panzhihua University, Panzhihua 617000, China

[4] Key Laboratory of Ministry of Education for Solid Waste Treatment and Resource Recycle, Southwest University of Science and Technology, Mianyang 621010, China

* Correspondence: xiaojunhui@swust.edu.cn; Tel.: +86-139-9019-0544

**Abstract:** In this study, a one-step rapid heating novel process was used to prepare high-purity anhydrous scandium chloride molten salt with low-purity scandium oxide. High-purity anhydrous ScCl₃ molten salt was used as the Sc-bearing raw material for preparing the Sc-bearing master alloy. Inert gas was used to enhance the purity of anhydrous scandium chloride and reduce the hydrolysis rate of scandium. The results show that high-purity scandium chloride (purity, 99.69%) with the scandium content of 29.61%, was obtained, and the hydrolysis rate of scandium was 1.19% under the conditions used: removing ammonium chloride; residual crystal water temperature of 400 °C; $m(Sc_2O_3):m(NH_4Cl)$ = 1:2.5; holding-time of 90 min; heating-rate of 12 °C/min; and argon flow of 7.5 L/min. XRD, SEM, and EPMA analyses further verified that anhydrous scandium chloride crystallization condition was relatively good and the purity of high-purity anhydrous scandium chloride approached the theory purity of anhydrous scandium chloride.

**Keywords:** scandium; scandium chloride; scandium oxide

## 1. Introduction

Scandium (Sc)—atomic number 21—lies in the periodic table of the fourth cycle III subgroup. Along with yttrium and lanthanide, scandium belongs to the rare earth family. It also has a particularity of mineralization with titanium, vanadium and iron that is not common with other rare earth elements. In the crust, the specific minerals of scandium are very rare. Moreover, there are only small economic reserves of Sc-bearing minerals including only thortveitite $(Sc,Y)_2Si_2O_7$, scandium phosphite $ScPO_4·2H_2O$, bazzite $Be_3(Sc,Al)_2Si_6O_{18}$, titanium silicate mineral $Sc(Nb,Ti,Si)_2O_5$ and befanamite $(Sc,Zr)_2Si_2O_7$, etc. In nature, scandium mostly occurs in ilmenite, zircon, bauxite, rare earth ore, ilmenite, V–Ti magnetite, tungsten ore, tin ore, uranium ore, coal and other minerals. In recent years, the combination of mineral processing and hydrometallurgical separation technology has been widely used in the field of comprehensive recovery and extraction of scandium, which greatly improves the concentration ratio and recovery rate of scandium. In future research and production, the recovery

process of main metals and other valuable metals should be considered—with the recovery and extraction process of scandium simplified, and the production cost reduced as much as possible [1–3].

Scandium is not only rare in the crust (only $7\times10^{-6}$ in the upper crust, equivalent to seven grams of scandium per ton of crust material), but also has a very low probability of forming scandium independent mineral in nature, which is only 0.4%. Current findings indicate that these rare independent minerals are mainly distributed in a few countries in Europe and Madagascar in Africa and have no real industrial value. In 1879, the Swedish chemist Neilson first discovered scandium in a silicon–beryllium yttrium mine and a black gold mine in Scandinavia in Europe. Scandium is normally dispersed in a number of minerals that contain other elements, like salt or sugar dissolved in water, and is not directly visible to the naked eye. Therefore, it is very difficult to obtain pure scandium metal. It took half a century from the discovery of scandium to the first successful extraction [4–6].

Scandium products are mainly used in Al–Sc alloy materials, scandium sodium halide high-intensity discharge (HID) lamps, SOFC, etc. Al–Sc alloys are mainly used in sporting goods production, automotive and aerospace industries. The higher power density of scandium zirconium-based solid electrolyte leads to the decrease of solid oxide fuel cell (SOFC) reaction temperature. Scandium lasers can be used in defense and medical fields. Scandium isotope is a common tracer in the oil refining industry. The International Adamas intelligence research center divides scandium's applications into three types: mature applications, emerging applications and future demand [7–10].

Scandium chloride can be used to make high melting-point alloys. The preparation of scandium intermediate alloys by doping scandium fluoride or scandium oxide metal thermal reduction and electrolysis has been reported. Scandium metal is prepared by mixing scandium metal into aluminum alloy. Scandium metal is expensive and has a large burning-loss during smelting. In the preparation of scandium fluoride by thermal reduction method, corrosion and toxic hydrogen chloride are used, and the thermal reduction temperature of the metal is also very high. The real yield of scandium by scandium oxide thermal reduction is not high. Organic solvent extraction is one of the main methods for separating and enriching scandium. In most cases, extraction is more efficient than precipitation or ion exchange, and can be used at all stages of scandium extraction. The solvent extraction method is characterized by a simple, easy-to-grasp method, rapid and good enrichment and separation effect and large processing capacity. However, these advantages require choosing a suitable extraction system and effectively considering all kinds of factors affecting the extraction separation in order to fully come into play. Extraction, washing and reverse extraction are three complementary processes in extraction and separation [11–14].

The deep research of scandium by Chinese scientists began with V–Ti magnetite in Panzhihua area of China. The value of scandium accounts for about 47% of Panzhihua V–Ti magnetite. The economic value of scandium is more than four hundred million USD, ranking first in the economic value of all elements of Panzhihua V–Ti magnetite. Because the grade of Panzhihua V–Ti magnetite is not high, it produces a large amount of tailings, which not only occupy much arable land, but also pollute the environment. There are rare scandium-containing minerals, making it difficult to separate and extract scandium. Consequently, scandium and its compounds become rather expensive. Scandium in V–Ti magnetite is mainly enriched in the tailings of titanium separation by beneficiation process, so it is necessary to extract scandium from the tailings of titanium separation. The tailings of V–Ti magnetite are pretreated by magnetic separation gravity separation electro separation process to enrich scandium to obtain scandium concentrate. To obtain scandium products with a high purity and recovery rate, it is necessary to combine multiple methods to improve the extraction process, to find more materials that can recover scandium—as well as improve the recovery and extraction methods of scandium, which will be of great significance to the effective use of the source [15–18].

Scandium oxide ($Sc_2O_3$) is the most common scandium compounds. Scandium oxide from high temperature burning is not soluble in dilute acid, but soluble in boiling concentrated nitric acid. Scandium chloride ($ScCl_3$) is an important raw material for the further preparation of scandium

metal and Al–Sc alloy [19–22]. To date, the key point of preparing $ScCl_3$ is how to reduce the hydrolysis rate of scandium chloride in $ScCl_3$. For this reason, a new one-step rapid heating process is proposed to prepare high-purity anhydrous $ScCl_3$ with the low-purity $Sc_2O_3$ (purity, 81.16%), along with the appropriate technological conditions. The high-purity anhydrous $ScCl_3$ will deeded as the important scandium-bearing raw materials for the preparation of Sc-bearing master alloy by aluminum–magnesium thermal reduction.

## 2. Materials and Methods

### 2.1. Sampling

The tailings samples of V–Ti magnetite from Panzhihua area of China were collected, which contained 38.61 g/t $Sc_2O_3$. The mineral composition of the tailings was complex and the content of scandium was very low. The scandium rough concentrate containing 76.98 g/t of $Sc_2O_3$ was further enriched by magnetic separation, gravity separation and electric separation from the tailing samples. The scandium oxide ($Sc_2O_3$, 81.16%) was obtained from the scandium rough concentrate by roasting, hydrolysis, hydrochloric acid leaching, extraction, purification, oxalic acid precipitation and roasting [23,24]. The main chemical composition analysis of scandium oxide is shown in Table 1.

**Table 1.** Chemical composition analysis results of scandium oxide (%).

| $Sc_2O_3$ | $Fe_2O_3$ | MnO | $SiO_2$ | $Al_2O_3$ | CaO | MgO | $K_2O$ | $Na_2O$ |
|---|---|---|---|---|---|---|---|---|
| 81.16 | 4.56 | 0.56 | 3.01 | 1.76 | 2.23 | 1.79 | 1.46 | 0.89 |

### 2.2. Chemical Reagent and Equipment

The main chemical reagents used in this test were ammonium chloride, ammonia, glacial acetic acid, anhydrous sodium acetate, ascorbic acid, hydrochloric acid ethylene diamine tetraacetic acid, zinc oxide, xylenol orange, Eriochrome Black T, nitrogen and argon with analytical purity were Tianjin chemical reagent research institute Co., Ltd., Tianjin China.

The main equipment used in the experiment included an electric atmosphere tube furnace (model: SX-6-16, Changsha Kehui Furnace Technology Co., Ltd., Changsha, China), high-temperature electric resistance furnace (≤1300 °C, Shanghai Shiyan Electric Furnace Co., Ltd., Shanghai, China), low-temperature electric resistance furnace (≤600 °C, Shanghai Shiyan Electric Furnace Co., Ltd. Shanghai, China), drying box (Shanghai Shiyan Yan Electric Furnace Co., Ltd., Shanghai, China), electro-thermostatic water bath, porcelain crucible (50 mL, 100 mL,150 mL, 200 mL), oscillator and vacuum filter (F 300, Southwest Chengdu Experimental Equipment Co., Ltd., Chengdu, China).

### 2.3. Experimental Design

In the preparation of high-purity anhydrous scandium chloride in the process, the main objective is to prevent hydrolysis of the scandium chloride generated (ScOCl). Owing to the high and low back, the ScOCl content influences the of scandium recovery rate. The preparation process of aluminum magnesium alloy among scandium also has adverse effect. Two kinds of dehydration methods to remove water contained in the crystallization of rare earth chlorides are vacuum dewatering and inert gases flow. The process of preparation and equation of scandium hydrolysis and combination reaction are shown in Equations (1) and (2):

$$ScCl_{3(s)} + H_2O_{(g)} \rightarrow ScOCl_{(s)} + 2HCl_{(g)} \tag{1}$$

$$ScOCl_{(s)} + 2NH_4Cl_{(s)} \overset{\Delta}{\rightarrow} ScCl_{3(g)} + 2NH_{3(g)} + H_2O_{(g)} \tag{2}$$

Scandium oxide is first dissolved by heating with concentrated hydrochloric acid. Then, the auxiliary salt and ammonium chloride are dissolved by heating with distilled water.

After mixing the two solutions, the mixture is filtered, and the fully stirred filtrate is put in an ordinary electric furnace for heating and evaporation. When a large number of crystals in the solution precipitate, the solution is transferred to a constant temperature water pot for heating, so that the water can evaporate as completely as possible. The solution is moved to a vacuum drying oven for vacuum evaporation and most of the crystallized water is removed. After the water evaporation of scandium chloride molten salt is complete, the molten salt block is taken out and ground into powder. Finally, the molten salt is put into a vacuum tube atmosphere furnace to remove ammonium chloride. The remaining crystal and the prepared anhydrous scandium chloride molten salt is stored in a dryer. By analyzing the content of scandium in molten salt and calculating the hydrolysis rate of scandium is determined. The calculation formula is shown in Equation (3). The process of preparing anhydrous scandium chloride is shown in Figure 1.

$$\text{Hydrolysis rate of scandium} = \left(S^T - Sc^S\right)/Sc^T \times 100\% \tag{3}$$

where $Sc^S$ is the soluble scandium content of anhydrous scandium chloride molten salt/% and $Sc^T$ is the scandium content of anhydrous scandium chloride molten salt/%.

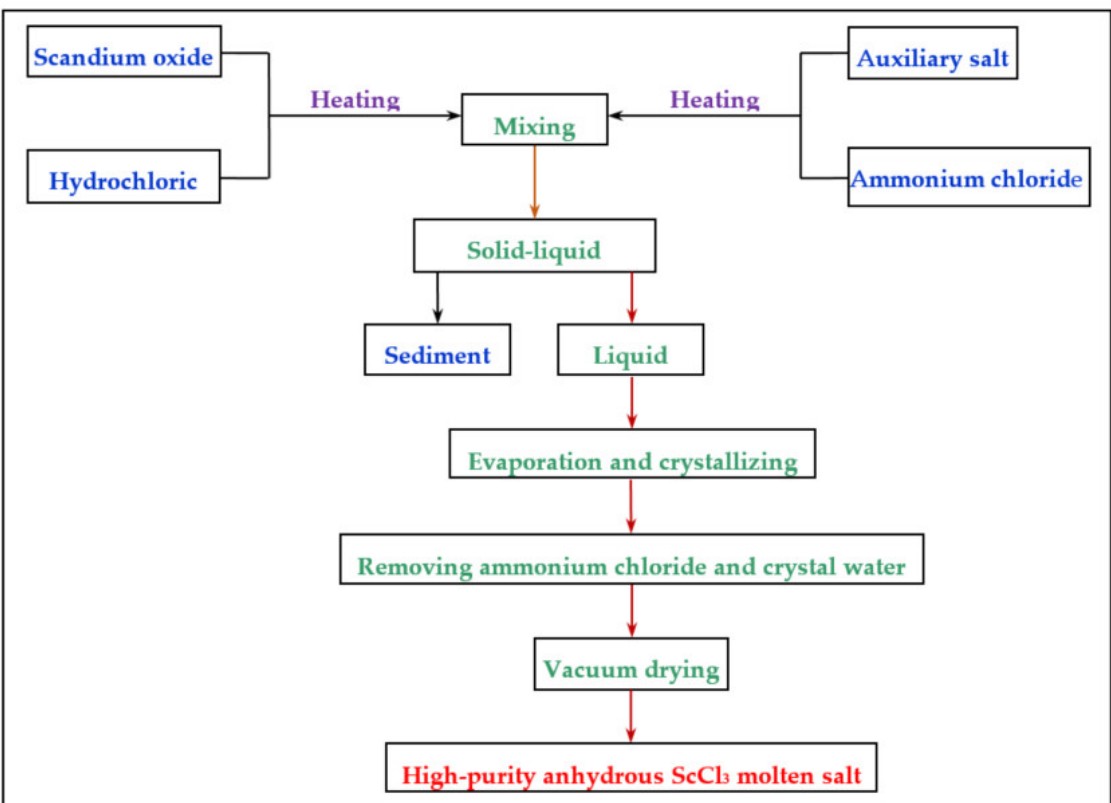

**Figure 1.** Scheme of preparing anhydrous scandium chloride molten salt.

The EDTA standard solution was prepared at a concentration of about 0.04mol/L. Then, the scandium content in scandium chloride salt was determined by EDTA complexometric titration. Scandium chloride oxide (ScOCl) formed by the hydrolysis of scandium chloride containing crystal water is insoluble in water, while scandium chloride is soluble in water. Therefore, the total scandium and soluble scandium contents in molten salt could be determined first.

The method for determination of the content of soluble scandium was: 0.5 g anhydrous scandium chloride molten salt samples were put in 100-mL beaker, 40 mL was added of distilled water to dissolve and fully mixed after filtration. Then, 10 mL pH material 4 drips of HAc—NaAc buffer solution was added the filtrate. Then, a small amount of ascorbic acid was added, with xylenol orange as an

indicator. Titration was completed with EDTA standard solution; the titration end solution changed from red to yellow. The content of soluble scandium in molten salt was calculated according to the consumption and concentration of EDTA standard solution and the weight of sample.

The total content of scandium was determined as follows: Another sample of 0.5 g molten salt was added to a 100-mL beaker. Then, 20 mL of melted concentrated hydrochloric acid was added. The remaining molten salt dissolved; all heating was stopped. Then, 30 mL of distilled water in a beaker was added with 1:1 ammonia to neutralize the acid solution. When a small amount of precipitation was seen, we stopped adding ammonia, and added 15 MLPH material 4 drips of HAc—NaAc buffer solution. A small amount of ascorbic acid was added, along with xylenol orange as an indicator. Titration was completed with an EDTA standard solution. The titration end solution changed from red to yellow. The total scandium content in molten salt was calculated according to the consumption and concentration of EDTA standard solution and the weight of sample.

### 2.4. Analysis and Characterization

The chemical composition of raw materials and anhydrous scandium chloride molten salt was analyzed by a Z-2000 atomic absorption spectrophotometer (Hitachi Co., Ltd. Tokyo, Japan).The phase composition of the anhydrous scandium chloride molten salt was analyzed by X-ray diffraction (XRD, X Pert pro, Panaco, The Netherlands).

The microstructure of the anhydrous scandium chloride molten salt was observed by SEM (S440, Hirschmann Laborgeräte GmbH & Co. KG, Eberstadt, Germany) equipped with an energy dispersive X-ray spectroscopy (EDS) detector (UItra55, Carl Zeiss NTS GmbH, Hirschmann Laborgeräte GmbH & Co. KG, Eberstadt, Germany).

The micro area composition of anhydrous scandium chloride molten salt was analyzed by electron probe microanalysis (EPMA-1720, Electron probe microanalyzer, Shimazu Instrument Company of Japan, Chengdu, China).

## 3. Results and Discussion

### 3.1. Effect of Temperature

Temperature is one of the key factors affecting the rate of scandium hydrolysis. If ammonium chloride and residual crystal water temperature is too slowly removed, ScOCl is formed—a hydrolytic product of scandium chloride that does not react well with ammonium chloride. If the temperature is too high, the sublimation speed of ammonium chloride is accelerated, and the degree of chlorination of ScOCl is reduced [25–28].

The results in Figure 2 show that the removing ammonium chloride and residual crystal temperature has an obvious hydrolysis rate of scandium. When the temperature was 400 °C, the hydrolysis rate of scandium presented a minimum value of 3.68% and scandium content of 23.44%. When the temperature was lower than 400 °C, the temperature increased and the hydrolysis rate of scandium decreased. When the temperature increased to 450 °C, the hydrolysis rate of scandium increased obviously. When the temperature was lower than 400 °C, the hydrolysis product ScOCl did not react completely with ammonium chloride, resulting in the increase of the hydrolysis rate of scandium. When the temperature was higher than 400 °C, the sublimation speed of ammonium chloride was accelerated, and the corresponding ScOCl could not be chlorinated completely.

Therefore, considering the removing ammonium chloride and surplus crystal water temperature of 400 °C was optimal. The anhydrous scandium chloride with scandium content of 23.44% was obtained, and the hydrolysis rate of scandium was 3.68%.

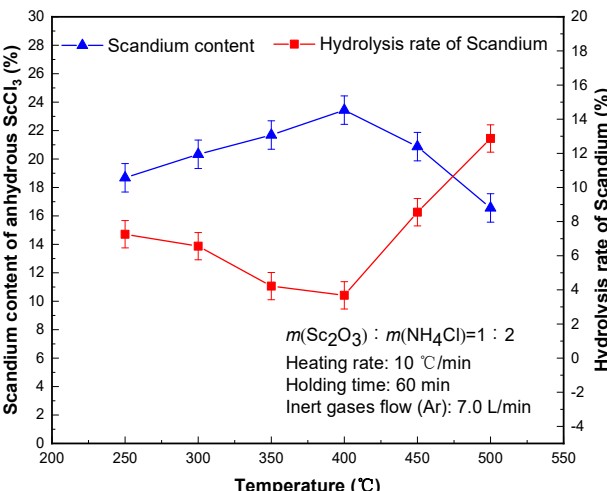

**Figure 2.** Effect of temperature on scandium content of anhydrous scandium chloride and hydrolysis rate of scandium.

### 3.2. Effect of Ammonium Chloride Dosage

The effect of different ammonium chloride dosage on scandium content of anhydrous scandium chloride and hydrolysis rate of scandium is shown in Figure 3. The amount of ammonium chloride also has a great influence on the hydrolysis rate of scandium. Too little or too much ammonium chloride may lead to incomplete chlorination of ScOCl or lead to increased reagent cost—which is not conducive to specific implementation [29–32].

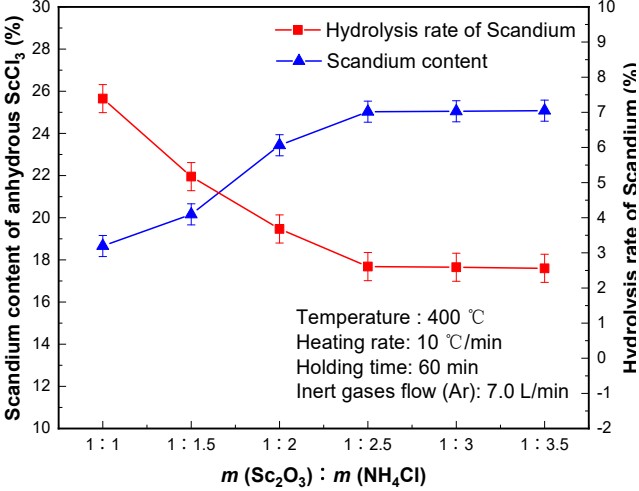

**Figure 3.** Effect of ammonium chloride dosage on scandium content of anhydrous scandium chloride and hydrolysis rate of scandium.

It can be seen from Figure 3 that with increasing ammonium chloride dosage, the hydrolysis rate of scandium decreased significantly, which is conducive to an increase of scandium content of anhydrous chlorinated molten salt. When $m(Sc_2O_3):m(NH_4Cl) = 1:2.5$, the hydrolysis rate of scandium decreased to 2.61%. When the dosage is increased to 1:3, the hydrolysis rate of scandium remained unchanged. Therefore, the weight ratio of ammonium chloride to scandium oxide in the raw material was 1:2.5. The anhydrous scandium chloride with scandium content of 25.03% was obtained, and the hydrolysis rate of scandium was 2.61%.

### 3.3. Effect of Holding Time

The effect of different holding times on the scandium content of anhydrous scandium chloride and hydrolysis rate of scandium is shown in Figure 4. Holding time is also one of the main factors affecting the scandium hydrolysis rate and scandium content of scandium chloride molten salt. If the insulation time is too short, ScOCl chloride is not complete, the hydrolysis rate of scandium is high, and ammonium chloride is not easy to be completely removed—seriously affecting the quality of scandium chloride molten salt and reducing the production efficiency [33–37].

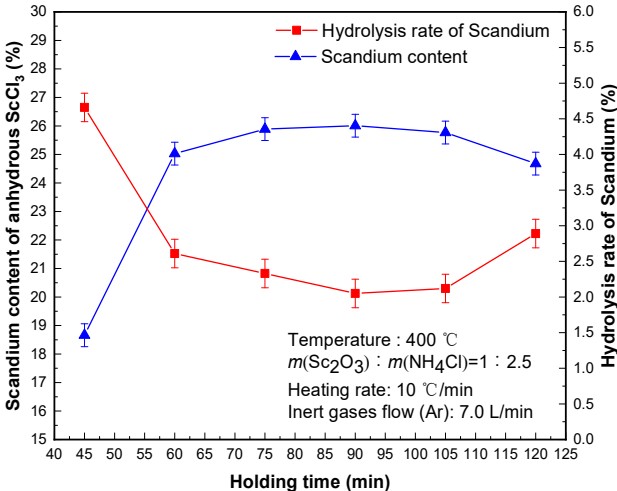

**Figure 4.** Effect of holding time on scandium content of anhydrous scandium chloride and hydrolysis rate of scandium.

The hydrolysis rate of scandium decreased significantly when the time was less than 75 min. When the time was extended to 90 min, the hydrolysis rate of scandium showed the lowest value of 2.05%. Because the time was too short, the chlorination of ScOCl was not complete; On the contrary, if the time was too long, the water vapor in the inert gas tended to hydrolyze scandium chloride due to the sublimation of ammonium chloride, leading to an increase of the hydrolysis rate of scandium. Therefore, 90 min was more suitable for the holding time of removing ammonium chloride and the residual crystal water. The anhydrous scandium chloride with scandium content of 26.01% was obtained, and the hydrolysis rate of scandium was 2.05%.

### 3.4. Effect of Heating Rate

Tests of the effect of different heating rate tests were carried out; the results are shown in Figure 5. Since the crystallization water in scandium chloride is removed under the protection of inert gases flow, there are strict requirements on the heating speed. The purpose is to make the crystal water rapidly gasify while the water vapor is quickly taken away by the inert gas—and to minimize the probability of water vapor hydrolyzing with scandium chloride.

When the heating rate was less than 12 °C/min, increasing the heating rate was conducive to reducing the hydrolysis rate of scandium. The rate of heating was 12 °C/min and the hydrolysis rate of scandium decreased to the minimum value of 1.86% and scandium content of 27.46% in anhydrous scandium chloride. When the heating rate increased to 16 °C/min, the hydrolysis rate of scandium increased to 2.34% instead. This was related to the factors such as the slow sublimation rate of the crystal water and the long action time of the water vapor and scandium chloride. Therefore, anhydrous scandium chloride with scandium content of 27.46% was produced and the hydrolysis rate of scandium was 1.86% under the condition used a heating rate of 12 °C/min.

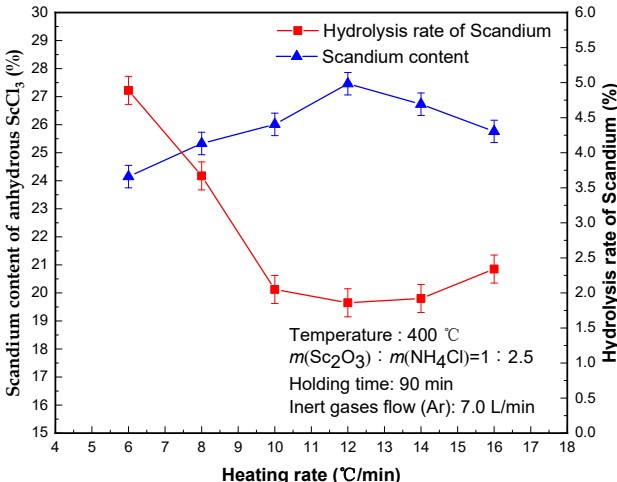

**Figure 5.** Effect of heating rate on scandium content of anhydrous scandium chloride and hydrolysis rate of scandium.

## 3.5. Effect of Inert Gases Flow

The main function of inert gas in the process of salt preparation is to protect the process atmosphere and the discharge of water vapor. If the flow rate of inert gas is too small, water vapor cannot be discharged in time. Too much flow will result in the waste of inert gas. The effect of different inert gases (argon and nitrogen) flow rate tests were conducted and the results are shown in Figure 6.

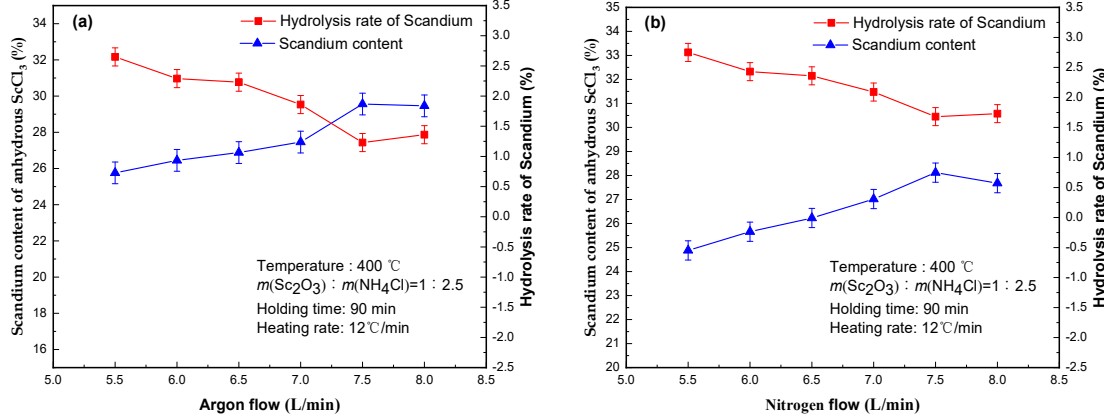

**Figure 6.** Effect of inert gases flow (**a**) argon; (**b**) nitrogen on scandium content of anhydrous scandium chloride and hydrolysis rate of scandium.

The test results in Figure 6 further confirm that under the same inert gas flow condition, anhydrous scandium chloride molten salt produced by argon as inert gas was superior to nitrogen as inert. The reason for this was that argon was heavier than air and could carry away water vapor more easily, which helps to improve the quality of molten salt. With the increase of inert gases flow rate, the hydrolysis rate of scandium first decreased and then increased. When the inert gases flow increased to 7.5 L/min, the hydrolysis rate of scandium decreased to 1.23% and the scandium of anhydrous scandium chloride increased to 29.56%. When the inert gases flow continued to increase, and the scandium hydrolysis rate changes little, which indicated that the inert gases flow was too small, the water vapor could not be discharged in time and the interaction time between scandium chloride and water vapor increased, which will lead to the increase of the scandium hydrolysis rate and the unnecessary argon was consumed. Therefore, argon flow rate of 7.5 L/min was more suitable.

### 3.6. Repeated Tests

In order to further validate test repeatability, repeated tests were carried out under the conditions employed as follows: a removing ammonium chloride and residual crystal water temperature of 400 °C; $m(Sc_2O_3):m(NH_4Cl) = 1:2.5$; a holding time of 90 min; a heating rate of 12 °C/min; and an argon flow of 7.5 L/min. The repeated results are shown in Tables 2 and 3.

**Table 2.** Repeated tests results of preparing anhydrous scandium chloride (%).

| Repeat | Scandium Content in Anhydrous ScCl₃ Molten Salt | Hydrolysis Rate of Scandium |
|---|---|---|
| 1 | 29.56 | 1.19 |
| 2 | 29.62 | 1.18 |
| 3 | 29.54 | 1.26 |
| 4 | 29.66 | 1.21 |
| 5 | 29.64 | 1.19 |
| 6 | 29.59 | 1.24 |
| 7 | 29.67 | 1.16 |
| 8 | 29.59 | 1.16 |
| Average | 29.61 | 1.19 |
| Range $(R = E_{\max} - E_{\min})$ | 0.13 | 0.100 |
| Arithmetic mean error $(\delta = \frac{\sum_{i=1}^{N}|d_i|}{N})$ | 0.039 | 0.027 |
| Sum square variation $(SS = \sum_{i=1}^{N} d_i^2 = \sum_{i=1}^{N}(E_i - \bar{E})^2)$ | 0.015 | 0.01 |
| Average deviation $(MS = \frac{SS}{f})$ | 0.002 | 0.001 |
| Standard deviation $(s = \sqrt{\frac{\sum(E_i - \bar{E})^2}{N-1}} = \sqrt{MS})$ | 0.045 | 0.032 |

**Table 3.** Main chemical composition analysis results of anhydrous scandium chloride (%).

| Sc | Cl⁻ | Fe₂O₃ | SiO₂ | Al₂O₃ | CaO | MgO | K₂O | Na₂O |
|---|---|---|---|---|---|---|---|---|
| 29.61 | 70.12 | 0.001 | 0.04 | 0.008 | 0.006 | 0.008 | 0.006 | 0.002 |

These results in Table 2 show that the anhydrous scandium chloride molten salt index (scandium content and hydrolysis rate of scandium) of repeated experiments was superior to the conditions of resulted. Scandium chloride with the scandium content of 29.61% was obtained, and hydrolysis rate of scandium was 1.19%.This indicated that it was feasible to prepare high-purity scandium chloride using non-high-purity scandium chloride raw materials by the process of rapid temperature desorption of ammonium chloride to a certain temperature and residual crystal water, and the hydrolysis rate of scandium chloride in scandium chloride was low. Compared with the pure scandium chloride molten salt, the scandium content in the prepared scandium chloride was 29.61%, which was close to the theoretical scandium content in scandium chloride was $45/(45 + 35.5 \times 3) \times 100\% = 29.703\%$, and the purity was $29.61\%/29.703 = 99.69\%$.

Meanwhile, it is known from the results in Table 3 reveal that the content of other impurities in the anhydrous scandium chloride molten salt also lower and provide high quality scandium-bearing raw materials for the subsequent preparation of aluminum–scandium alloy.

*3.7. Analysis and Characterization of Anhydrous Scandium Chloride Molten Salt*

A novel one-step rapid heating process was used to prepare preparing high-purity anhydrous scandium chloride with non-high-purity scandium oxide, and the product index is ideal. X-ray diffraction (XRD), scanning electron microscopy (SEM), energy dispersive spectroscopy (EDS), electronic probe microanalyzer (EPMA) were used to analyze micro area composition of the high-purity anhydrous scandium chloride molten salt. The XRD analysis results of high-purity anhydrous scandium chloride were show in Figure 7. The SEM-EDS images analysis results of the high-purity anhydrous scandium chloride are shown in Figure 8. The micro area composition analysis results of high-purity anhydrous scandium chloride are shown in Table 4.

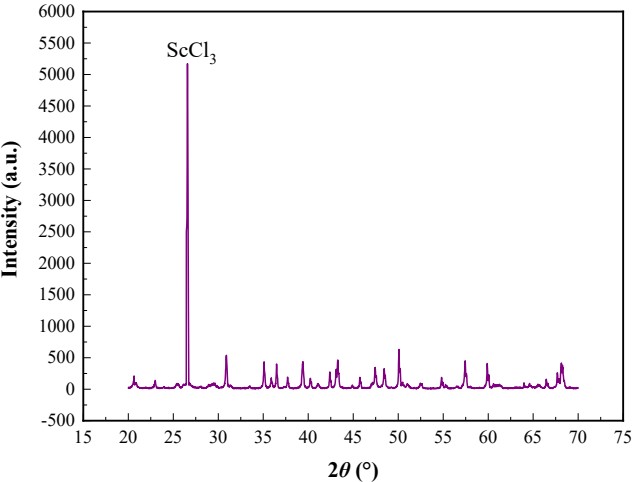

**Figure 7.** XRD diffractograms of high-purity anhydrous scandium chloride molten salt.

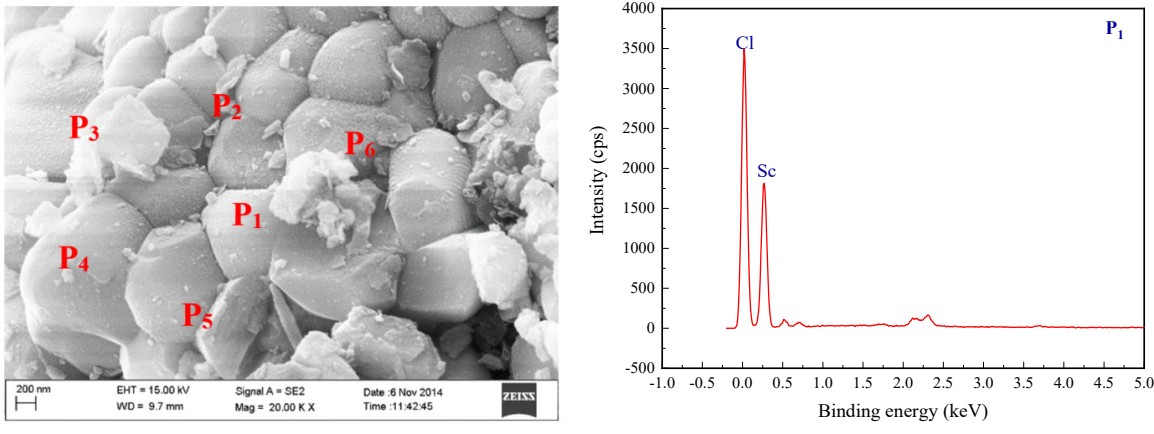

**Figure 8.** SEM-EDS images analysis results of high-purity anhydrous scandium chloride molten salt.

The XDR, SEM-EDS and EPMA analysis resulted of anhydrous scandium chloride confirm that the crystallization condition of high-purity anhydrous scandium chloride was relatively good, different location of the electron probe micro elements also show that the impurity element content was lower and close to the theoretical purity of scandium chloride molten salt. Furthermore, further illustrate that preparing high-purity anhydrous scandium chloride with high-purity scandium oxide was feasible using the process of inert gases flow step in quick heating and removal of ammonium chloride and residual crystal water.

**Table 4.** Electron probe microanalysis (EPMA) micro area composition of high-purity anhydrous scandium chloride (%).

| Position | Sc | Cl | Si | Ca | Al | Totals |
|:--------:|:------:|:------:|:-----:|:-----:|:-----:|:------:|
| $P_1$ | 29.589 | 70.156 | 0.005 | 0.002 | 0.012 | 99.764 |
| $P_2$ | 29.662 | 70.231 | 0.006 | 0.003 | 0.013 | 99.915 |
| $P_3$ | 29.617 | 70.119 | 0.003 | 0.004 | 0.024 | 99.767 |
| $P_4$ | 29.622 | 70.117 | 0.006 | 0.003 | 0.031 | 99.779 |
| $P_5$ | 29.599 | 70.205 | 0.009 | 0.006 | 0.043 | 99.862 |
| $P_6$ | 29.669 | 70.098 | 0.008 | 0.005 | 0.039 | 99.819 |
| Average | 29.626 | 70.154 | 0.006 | 0.004 | 0.027 | 99.818 |

## 4. Conclusions

Based on the test results of preparing high-purity anhydrous $sccl_3$ molten salt obtained in this study, we drew the main following conclusions:

(1) High-purity anhydrous $ScCl_3$ molten salt was prepared using one-step rapid heating process to a certain temperature in an inert gas stream to remove ammonium chloride and residual crystal water. High-purity scandium chloride molten salt (purity, 99.69%) with the scandium content of 29.61%, was obtained and the hydrolysis rate of scandium was 1.19% under the conditions used: a removing ammonium chloride and residual crystal water temperature of 400 °C; $m(Sc_2O_3):m(NH_4Cl) = 1:2.5$; a holding time of 90 min; a heating rate of 12 °C/min; and an argon flow of 7.5 L/min. The index of high-purity scandium chloride molten salt was ideal.

(2) XRD, SEM and EPMA analysis results also further showed that anhydrous scandium chloride crystallization condition is relatively good. Impurity elements are lower in high-purity anhydrous $ScCl_3$ molten salt. The purity of scandium chloride are close to the theory purity of scandium chloride. Therefore, it is feasible to prepare the high-purity anhydrous scandium chloride by the quick heating a certain temperature in inert gases flow process with non-high-purity scandium oxide. The high-purity scandium chloride molten salt can be used as an important raw material for preparing Al–Sc master alloy using Al–Mg thermoreduction method.

**Author Contributions:** This is a joint work of the seven authors; each author was in charge of their expertise and capability: J.X.: writing, formal analysis and original draft preparation; W.D. and C.C.: conceptualization; Y.P.: validation; T.C. and K.Z.: methodology; Z.Z.: investigation. All authors have read and agreed to the published version of the manuscript.

**Funding:** This work was supported by the Sichuan Science and Technology Program (Grant Nos.2018FZ0092, Nos.201FS0451 and Nos.201FS0452); China Geological Big Survey (Grant No. DD20190694); Key Laboratory of Sichuan Province for Comprehensive Utilization of Vanadium and Titanium Resources Foundation (2018FTSZ35).

**Conflicts of Interest:** The authors declare no conflict of interest. The funders had no role in the design, analyses and interpretation of any data of the study.

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
