# Peer review of "Preparing High-Purity Anhydrous ScCl3 Molten Salt Using One-Step Rapid Heating Process"

_applsci, doi:10.3390/app10155174_

Round 1

Reviewer 1 Report

The manuscript describes a study on the conversion of scandium sesquioxide into scandium trichloride, with a view to avoiding the hydrolysis of the chloride formed. It is acknowledged that research on the various aspects of scandium metallurgy is a timely subject, given scandium’s unique role in many alloys and ceramics. Unfortunately however, the manuscript cannot be recommended for publication in its present form. Thorough restructuring and rewriting are mandatory.

  1. The Introduction is far too long, repetitive and unstructured. Below is one particularly striking example for each of these statement while there are plenty more.

It contains information that one would expect to find in textbooks but not in technical papers, for instance, “Pure scandium is silver-white with a yellowish tint of metallic sheen and is quite soft.”

The high reactivity of scandium with water and oxygen is mentioned several times.

History, mining and applications should be discussed in separate paragraphs, not in a random melange.

On the scientific side, there is clearly something wrong with the melting points quoted for Re, Mo and Co. Has anyone ever proofread this manuscript?

  1. The experimental part is written like a lab report and virtually incomprehensible, owing to lack of structure and excessively long sentences.

Chapters 2.2 and 2.3 are just a collection of chemicals and pieces of equipment. In a technical paper one would expect to find information on what was done which each chemical and with what type of equipment.

The experiments need to be described in a clear and concise manner. A single sentence of the type “Scandium oxide first heated with concentrated hydrochloric acid to dissolve, and then dissolve salt and NH4Cl using distilled water heating, after mixing the two solution, to filter the mixture, the filtrate after fully mixing on the ordinary electric furnace heating evaporation, when there are a large number of crystal to stay a solution to the constant temperature water soluble heating pan, make the moisture to evaporate as far as possible fully, and then remove the solution to vacuum evaporation in the vacuum drying oven, take off a part of the water of crystallization, stay scandium chloride in the water evaporation after completely, pieces and ground into a powder form, finally put in the vacuum tube type atmosphere in the furnace ammonium chloride and residual water of crystallization, anhydrous scandium chloride system, anhydrous scandium chloride put in dryer” is unacceptable.

As a specific point, can the Authors check that the manufacturer of the X-ray diffractometer is indeed called “Panaco” rather than PANalytical/Philips”.

  1. The results and discussion part is presented slightly better than the foregoing chapters. However, with the experimental part effectively incomprehensible, it is difficult to appreciate the results because one can only guess what exactly was done to arrive at the various graphs and numbers.

For example, what is a “residual crystal temperature”? The term has not been introduced before.

Or what is meant by “When the temperature is lower than 400℃, the temperature increases…”?

Or by “ScOCl chloride is not complete…”?

Or “the index of repeated experiments is superior to the conditions of results”? And what is that “product index”?

  1. The quality of the English is absolutely insufficient for a journal with international ambitions. In many cases it is impossible to understand what shall be communicated. The Authors must resort to professional support!

Author Response

Response to Reviewer 1 comments

Point 1: 1. The Introduction is far too long, repetitive and unstructured. Below is one particularly striking example for each of these statement while there are plenty more.

It contains information that one would expect to find in textbooks but not in technical papers, for instance, “Pure scandium is silver-white with a yellowish tint of metallic sheen and is quite soft.”

The high reactivity of scandium with water and oxygen is mentioned several times.

History, mining and applications should be discussed in separate paragraphs, not in a random melange.

On the scientific side, there is clearly something wrong with the melting points quoted for Re, Mo and Co. Has anyone ever proofread this manuscript?

Response 1: Thank you for your careful review. We have revised and corrected the content of this part. In addition, this refers to the boiling point of metal, so we have reviewed the relevant literature again, please review it again.

Point 2: 2. The experimental part is written like a lab report and virtually incomprehensible, owing to lack of structure and excessively long sentences.

Chapters 2.2 and 2.3 are just a collection of chemicals and pieces of equipment. In a technical paper one would expect to find information on what was done which each chemical and with what type of equipment.

The experiments need to be described in a clear and concise manner. A single sentence of the type “Scandium oxide first heated with concentrated hydrochloric acid to dissolve, and then dissolve salt and NH4Cl using distilled water heating, after mixing the two solution, to filter the mixture, the filtrate after fully mixing on the ordinary electric furnace heating evaporation, when there are a large number of crystal to stay a solution to the constant temperature water soluble heating pan, make the moisture to evaporate as far as possible fully, and then remove the solution to vacuum evaporation in the vacuum drying oven, take off a part of the water of crystallization, stay scandium chloride in the water evaporation after completely, pieces and ground into a powder form, finally put in the vacuum tube type atmosphere in the furnace ammonium chloride and residual water of crystallization, anhydrous scandium chloride system, anhydrous scandium chloride put in dryer” is unacceptable.

As a specific point, can the Authors check that the manufacturer of the X-ray diffractometer is indeed called “Panaco” rather than PANalytical/Philips”.

Response 2: Thank you for your precious suggestion. We have adjusted the content of this part. Please review it.

Point 3: 3.The results and discussion part is presented slightly better than the foregoing chapters. However, with the experimental part effectively incomprehensible, it is difficult to appreciate the results because one can only guess what exactly was done to arrive at the various graphs and numbers.

For example, what is a “residual crystal temperature”? The term has not been introduced before.

Or what is meant by “When the temperature is lower than 400℃, the temperature increases…”?

Or by “ScOCl chloride is not complete…”?

Or “the index of repeated experiments is superior to the conditions of results”? And what is that “product index”?

Response 3: Temperature is the temperature at which the remaining ammonium chloride and the remaining crystal water are removed. It is true that there is an improper statement here, which makes it difficult for people to understand. According to our experimental results, we consulted a large number of literature materials and modified the content of this part.

Point 4: The quality of the English is absolutely insufficient for a journal with international ambitions. In many cases it is impossible to understand what shall be communicated. The Authors must resort to professional support!

Response 4: Thank you for your valuable suggestions. We have made a lot of amendments to the language problem of the article, please review it.

Finally, I sincerely thank you for putting forward such precious suggestions for revision of our research. We have gained a lot, especially providing important guidance for our follow-up research work. There may still be insufficient support for this article. Please give us your valuable suggestions again. We are still very grateful. If you have any questions, please feel free to contact us. Wish you a healthy, happy and all the best in 2020.

Yours sincerely,

Junhui Xiao

Reviewer 2 Report

The article entitles „Preparing High-Purity Anhydrous ScCl3 Molten Salt Using One-Step Rapid Heating Process” authored by Xiao Junhui, Chao Chen, Wei Ding, Yang Peng, Kai Zou, Tao Chen, Zhiwei Zou presents interesting concepts and methods for prepare high-purity anhydrous scandium chloride molten salt with scandium oxide.

All the used methods do not raise any objections. The research methodology is well thought out and carried out in accordance with applicable standards. The test results and conclusions are correctly described.

This manuscript is interesting and presents new information but some issues should be explained and corrected before publications.

The equation of reaction 1 is unbalanced.

This reaction 1 is questionable. The authors do not confirm that they have received scandium chloride oxide. It is well known that ScOCl3 is formed by heating ScCl3 in the air at 700°C. In this eq. should be ScOCl. Did the authors use hydrate or anhydrous scandium chloride in reaction 1?

The description of the methodology should be checked.

The authors cite item 19 as a reference to the method of obtaining the scandium oxide. This reference is about titanium.

Author Response

Response to Reviewer 2 comments

Point 1: The equation of reaction 1 is unbalanced.

Response 1: Thank you very much for carefully reviewing the work. We have modified it. Please review it.

Point 2: This reaction 1 is questionable. The authors do not confirm that they have received scandium chloride oxide. It is well known that ScOCl3 is formed by heating ScCl3 in the air at 700°C. In this eq. should be ScOCl. Did the authors use hydrate or anhydrous scandium chloride in reaction 1?

Response 2:  There is something wrong with the chemical reaction formula here. We reviewed the relevant literature and made some modifications.

Point 3: The description of the methodology should be checked.

Response 3: Thank you for your suggestion. We revised the content again and made some modifications. Thank you.

Point 4: The authors cite item 19 as a reference to the method of obtaining the scandium oxide. This reference is about titanium.

Response 4: Here is a mistake in our work. We conform to the literature again and adjust the references. Please review it. Thank you.

Thank you very much for taking your precious time to review the article in your busy work, and thank you for your recognition of our research work. We will revise the article according to the opinions of other reviewers. Please review it again. I hope our modification work can get your approval again, and looking forward to your good news! 

Kind regards,

Junhui Xiao

Round 2

Reviewer 2 Report

The authors modified the manuscript accordingly to the comments. I  recommend the revised version to be accepted for publication in Applied Sciences. In my opinion, the amended article may be approved for publication in the present form.